# Evaluation and Discrimination of Lipid Components and Iron and Zinc Levels in Chicken and Quail Eggs Available on the Polish Market

**DOI:** 10.3390/foods13101571

**Published:** 2024-05-17

**Authors:** Małgorzata Czerwonka, Agnieszka Białek, Dorota Skrajnowska, Barbara Bobrowska-Korczak

**Affiliations:** 1School of Health and Medical Sciences, University of Economics and Human Sciences in Warsaw, Okopowa 59, 01-043 Warsaw, Poland; a.bialek@vizja.pl; 2Department of Toxicology and Food Science, Faculty of Pharmacy, Medical University of Warsaw, Banacha 1, 02-097 Warsaw, Poland; dorota.skrajnowska@wum.edu.pl (D.S.); barbara.bobrowska@wum.edu.pl (B.B.-K.); 3The Kielanowski Institute of Animal Physiology and Nutrition, Polish Academy of Sciences, Instytucka 3, 05-110 Jabłonna, Poland

**Keywords:** chicken eggs, quail eggs, fatty acids, iron, zinc, cholesterol

## Abstract

All over the world, birds’ eggs are an important and valuable component of the human diet. This study aimed to compare the content of lipid components and their nutritional value as well as iron and zinc levels in chicken and quail eggs commonly available on the market. In egg lipids, unsaturated fatty acids were dominant, especially oleic acid, the content of which was about 40% of the total fatty acids (TFAs). Linoleic acid was the major polyunsaturated fatty acid. Compared to other products of animal origin, eggs were characterized by favorable values of lipid quality indices, especially the index of atherogenicity, thrombogenicity, and the hypocholesterolemic-to-hypercholesterolemic ratio. In the present study, no differences were found in the content of tested nutrients between eggs from different production methods (organic, free-range, barn, cages). Based on linear discriminant analysis, inter-breed differences were noticed. Cluster analysis showed that eggs enriched in n3 PUFAs (according to the producers’ declarations) differed from other groups of chicken eggs. However, in eggs from one producer only, the amount of EPA and DHA exceeds 80 mg per 100 g, entitling the use of the nutrition claim on the package. Quail eggs differed from chicken eggs in FA profile and cholesterol and iron levels.

## 1. Introduction

Bird eggs have always been an important and valuable component of the human diet. The consumption of eggs depends on various factors such as economics, culture, religion, and eating habits. However, all over the world, they are perceived as nutritious, easily digestible, and delicious meal ingredients [1,2].

In the 1960s, it was recommended to limit the consumption of eggs due to the high content of cholesterol, which was supposed to promote hypercholesterolemia in the body [3]. This view became deeply ingrained in consumers’ thoughts. Today, it is known that dietary cholesterol does not affect the levels of low-density lipoproteins in the body. The greatest hypercholesterolemic/atherogenic effect is related to saturated fatty acids [4]. The unsaturated fatty acids that dominate egg fat have the opposite effect. Furthermore, recent studies on egg consumption demonstrate that eating those products daily does not alter LDL levels and can improve postmeal metabolic responses [5]. The profile of fatty acids in the diet has a significant impact on the body’s lipid balance. In addition to fatty acids and cholesterol, the lipids of eggs include choline, fat-soluble vitamins, and carotenoids [6]. A whole egg contains many valuable nutrients and bioactive ingredients: wholesome proteins, most vitamins, and minerals, including iron and zinc [7]. Eggs are a great alternative to meat in the diet.

There are several product groups in the egg market. The most popular is the division of eggs obtained from various species of birds. All over the world, chicken eggs (*Gallus domesticus*) are the most popular, but eggs of quails, ducks, turkeys, and other species are also present on the market [8]. In the European Union, hen eggs are classified in terms of weight and production method. Organic eggs have the number 0 marked on the shell, free-range eggs have the number 1, barn eggs 2, and cage eggs 3 [9]. There are also premium products on the market: eggs from less common breeds (e.g., Green-legged Partridge) or enriched with selected vitamins, minerals, or other bioactive ingredients [10]. Studies have shown that the nutritional value of birds’ eggs depends on many factors, such as the species and breed of animals from which the eggs were obtained, the production method, and the type of feed [6]. Most of the research, however, was conducted under controlled conditions. There are much fewer studies that have evaluated market products.

This study aimed to compare the content of lipid components as well as iron and zinc levels in chicken and quail eggs commonly available on the market. Because the amount of lipids in egg whites usually does not exceed 0.2% [8], this study was limited to the nutritional composition of egg yolks.

## 2. Materials and Methods

### 2.1. Research Material

The research material comprised six groups of chicken eggs and quail eggs available on the Polish market. The groups were as follows:▪Chicken eggs from organic production (E);▪Free-range chicken eggs (F);▪Barn chicken eggs (B);▪Chicken eggs from caged hens (C);▪Chicken eggs with an increased content of n3 fatty acids (N3) (according to producers’ declarations/nutrition claims on the package);▪Chicken eggs from Green-legged Partridge (GL);▪Partridge quail eggs (Q).

The packaging of the eggs from the first five groups lacked an indication of the breed of the hens from which the eggs were obtained. Laying hen breeds predominating in the region are Rhode Island Red and Leghorn. The characteristics of examined egg samples and the weight parameters of all groups of eggs are presented in Table 1. In each group, 8 brands (producers) were evaluated. Three eggs were randomly collected from each package (brand) and separate analyses were performed for each egg (*n* = 3 × 8 = 24 per group). Six eggs were collected from each quail egg pack. Due to their small size, three yolks were combined into one sample (*n* = 6/3 × 8 = 16).

### 2.2. Analytical Methods

#### 2.2.1. Fat Content Determination

Fat content in egg yolk samples was determined gravimetrically after performing extraction three times with a mixture of chloroform/methanol (*v*/*v* 2:1) and solvent evaporation under a stream of nitrogen, according to the procedure described by Folch et al. [11].

#### 2.2.2. Fatty Acid Analysis

The fatty acid content and profile were determined by gas chromatography with a flame ionization detector after a methylation procedure based on the method described by Białek et al. [12]. To extract egg yolk fat (about 20 mg), 1 mL of 0.5 M NaOH in methanol was added and heated at 80 °C for 15 min. Then, 1 mL of BF_3_ solution in methanol (14% *w*/*v*) was added and again heated at 80 °C for 15 min. Then, extraction of FAME was performed by adding 1 mL of a saturated solution of NaCl in water and 1 mL of hexane and shaking. After phase separation, the hexane layer was transferred to a 2 mL vial and injected into the column. 

Analyses were performed on a gas chromatograph with a flame ionization detector (Shimadzu GC-17A, Kyoto, Japan). Chromatographic separations were conducted on a capillary column SGE BPX70 (60 m/0.25 mm ID/film thickness 0.20 μm; Ringwood, Australia). Helium was used as the carrier gas (flow: linear velocity at 0.9 mL min^−1^), the injection was 1 μL, and the split was set to 10. The injector was heated to 250 °C and the detector to 270 °C. The temperature program was as follows: initial temperature—140 °C for 1 min, increase by 20 °C per min to 200 °C, hold for 20 min, increase by 5 °C per min to 220 °C, hold for 25 min. FAME standards (Supelco 37Component FAME Mix, Sigma, St. Louis, MO, USA) were used to identify the FAs present in the samples. 

Based on the percentage of fatty acids, the following indices of lipid quality were calculated [13,14]:▪Flesh-lipid quality (FLQ):FLQ = EPA + DHA▪Index of atherogenicity (AI):AI = [(4 × C14:0) + C16:0]/(MUFA + n3 PUFA + n6 PUFA)▪Index of thrombogenicity (TI):TI = [C14:0 + C16:0 + C18:0]/[(0.5 × MUFA) + (0.5 × n6 PUFA) + (3 × n3 PUFA) + (n3 PUFA/n6 PUFA)]▪Hypercholesterolemic fatty acids (OFAs):OFA = C14:0 + C16:0▪Hypocholesterolemic fatty acids (DFAs):DFA = C18:0 + MUFA + PUFA▪Hypocholesterolemic/hypercholesterolemic ratio (H/H):H/H = (c9 C18:1 OL + C18:2 LA + C18:3 ALA)/(C14:0 + C16:0)


In the above formulae, EPA—eicosapentaenoic acid (c5c8c11c14c17 C20:5, EPA); DHA—docosahexaenoic acid (c4c7c10c13c16c19 C22:6); MUFAs—monounsaturated fatty acids; PUFAs—polyunsaturated fatty acids; n3 PUFAs—polyunsaturated fatty acids of the n3 family; n6 PUFAs—polyunsaturated fatty acids of the n6 family; C14:0—myristic acid; C16:0—palmitic acid; C18:0—stearic acid; c9 C18:1 OL—oleic acid; C18:2 LA—linoleic acid (c9c12 C18:2); C18:3 ALA—α-linolenic acid (c9c12c15 C18:3).

#### 2.2.3. Cholesterol Content Determination

Cholesterol was determined using the RP-HPLC method with UV detection at 210 nm. To about 50 mg of egg yolk, 2 mL 0.5 M KOH in ethanol and 20 µL of butylated hydroxytoluene solution (5 mg mL^−1^ in ethanol) were added. The sample was put into an ultrasonic bath for 20 min and heated at 80 °C for 30 min. Then, 4 mL of the citric acid solution (4.5% *w*/*v* in water) and 4 mL of hexane were added and shaken for 5 min. After phase separation, the upper layer was transferred into a 10 mL volumetric flask. Extraction was repeated twice with 2 mL hexane. A volumetric flask was filled to the mark with hexane and mixed thoroughly. A 500 μL volume of the solution was transferred to a 2 mL vial and evaporated under a stream of nitrogen. The dry residue was dissolved in 500 μL of isopropanol. 

Chromatographic analysis was performed on a Merck Hitachi HPLC system (Darmstadt, Germany; pump: L-7100; UV–VIS detector: L-7420). Separation was carried out on Luna 5uC18(2) (Phenomenex, Torrance, CA, USA; pore size: 100 Ĺ, L × I.D.: 150 × 2 mm) operated at 35 °C. Isocratic elution was executed with a mixture of acetonitrile and isopropanol (9:1, *v*/*v*). A flow rate of 0.4 mL min^−1^ and an injection volume of 10 μL were used. The cholesterol concentration was calculated against the calibration curve.

#### 2.2.4. Iron and Zinc Content Determination

Iron and zinc determination was carried out by atomic absorption spectrometry. Before the analysis, the egg yolk samples were mineralized by a microwave mineralizer (Plazmatronika, Ertec, Wroclaw, Poland) in a nitric acid medium. Then, 0.3–0.4 g of sample was weighed directly into a closed PTFE vessel and 6 mL of nitric acid (ultra-pure) was added. The heating program was performed in three steps: [a] 4 min, power: 80%, pressure: 19–22 atm; [b] 4 min, power: 90%, pressure: 23–26 atm; [c] 8 min, power: 100%, pressure 33–36 atm. The mineralizate was diluted with ultra-pure water to 10 mL. 

The determination was carried out by an air–acetylene flame atomic absorption spectrometer (Philips Analytical PU-9100, Philips, Cambridge, United Kingdom) with a single-element hollow cathode lamp. The analytical wavelength was 248.3 nm for Fe and 213.9 nm for Zn. Mineral concentrations were calculated against the calibration curve.

### 2.3. Statistical Analysis

All chemical analyses were performed in triplicate. The results are presented as mean values ($\bar{x}$) ± standard deviation (SD). Distributions of the data (normality) were assessed by the Shapiro–Wilk test. Differences among examined groups of eggs were analyzed with a one-way ANOVA (α = 0.05), with a post hoc RIR Tukey test (α = 0.05). A cluster analysis of lipid component content (fatty acids, cholesterol) in egg yolks was performed; the Ward agglomeration procedure and the Euclidean function of the distance were applied. A cut-off point was established at 33% of the maximum distance, according to Sneath’s criterion. To evaluate whether fatty acid and cholesterol contents in eggs significantly differed among groups, principal component analysis (PCA) was performed. A matrix of 32 variables was used. The number of principal components (PCs) was chosen by the screen test criterion. To obtain appropriate rules for classifying egg samples into experimental groups, a linear discriminant analysis (LDA) for examined variables (fatty acid and cholesterol contents) differing significantly among clusters was performed. Relevant discriminant functions were calculated in a stepwise progressive method, with the adopted tolerance value 1 − R2 = 0.01 to optimize the LDA. A correlation analysis was performed to assess the relationship between the content of individual lipid components and iron and zinc.

All results were evaluated using Statistica 13.3 software (StatSoft, Kraków, Poland).

## 3. Results

In the present study, the content of lipid components (fatty acids, cholesterol), iron, and zinc in selected groups of eggs available on the market was assessed. Hen egg weight, shell weight, yolk weight, and the share of edible parts between products from hens did not differ significantly. Quail eggs were clearly smaller; therefore, the other factors mentioned above were also much lower. Fat content in egg yolks differed between egg groups: the largest was in products from ecological production (32.0%), and the lowest was in egg yolks from caged hens (25.7%). The characteristics of examined egg samples are presented in Table 1.

The examined groups of eggs differed in the content and profile of fatty acids. The fatty acid and cholesterol contents in egg yolks (in mg g^−1^ of yolk) are presented in Table 2, while the share of the main groups of fatty acids in the total FA content and FA quality indices are in Table 3.

The content of saturated fatty acids (SFAs) ranged from 32.8% (N3) to 36.2% (Q) of the total fatty acids (TFA) content. The dominant saturated fatty acid was palmitic acid; its average content was 25.1% TFA. The lowest C16:0 levels were found in egg yolks from caged hens (24.2% TFA), and the highest in barn (25.8% TFA) and quail eggs (25.5% TFA). The second SFA in egg yolks was stearic acid (C18:0). The highest levels were determined in quail eggs (9.1% TFA). The mean content of this acid in chicken eggs was 7.6% TFA. The content of remaining SFAs (C13:0, C14:0, C15:0, C17:0, C20:0, C21:0, C24:0) determined in eggs did not exceed 0.5% TFA.

Monounsaturated fatty acids dominated in egg yolks, which constituted on average 45.7% of the TFAs. Oleic acid was the main one. Although its content significantly differed between the examined eggs, the share in the total pool of fatty acids (average 39.7% TFA) was similar (*p* = 0.189). The content of palmitoleic acid (C16:1) ranged from 3.04 (C) to 4.13 (Q)% TFA, while the c11 C18:1 acid ranged from 2.08 (Q) to 3.27 (GL)% TFA. The content of the remaining MUFA (C14:1, C15:1, C17:1, C20:1) did not exceed 0.2% TFA.

The content of polyunsaturated fatty acids had an average of 19.7% TFA and did not differ significantly between the examined groups. Linoleic acid was the dominant PUFA. Its share [%] in TFAs ranged from 14.4 (Q) to 17.4 (C). The second-largest n6 PUFA was arachidonic acid (c5c8c11c14C20:4, AA); its amount was between 1.58 (N3) and 1.96 (Q)% TFA. The content of α-linolenic acid, belonging to the n3 family, did not differ significantly between the groups; the average level was 0.72%. The content of long-chain PUFAs from the n3 fatty acid family was low. The share of eicosapentaenoic acid in any group did not exceed 0.05% TFA. The docosahexaenoic acid (c4c7c10c13c16c19 C22:6, DHA) level was higher, ranging between 0.56 (F) and 0.8 (E, Q) % of TFA. The FLQ index, being the sum of the share of EPA and DHA in TFA, differed significantly between the studied groups of products. The highest value was recorded for N3 and quail eggs (0.86) and the lowest for free-range chicken eggs (0.57). The content of the remaining determined PUFAs (C18:3 n6, C20:2, C20:3, C22:2) was low, not exceeding 0.2% of TFA. The n6-to-n3 polyunsaturated fatty acid ratio fluctuated within a wide range between the studied groups. Egg yolks from free-range chickens had the highest value (24.2), while egg yolks from ecological production and quail eggs had the lowest (10.6).

The index of atherogenicity (AI) was calculated between 0.39 (E, C, N3) and 0.44 for quail egg fat. The index of thrombogenicity (TI) was the lowest for N3 egg fat (0.85) and the highest for quail egg fat (0.99). The richest in hypercholesterolemic fatty acids (OFAs) was barn chicken eggs. OFA was lowest in N3 egg fat. Hypocholesterolemic fatty acid (DFA) levels are inversely proportional to OFAs (r = 0.989). DFAs were highest for fat from N3 eggs (74.36) and the lowest for fat from barn chicken eggs (72.6). The hypocholesterolemic-to-hypercholesterolemic ratio ranged between 2.04 (Q) and 2.33 (C, N3). 

Cholesterol levels ranged from 13.1 mg g^−1^ to 14.4 mg g^−1^. The highest content of this compound was found in barn chicken egg yolks and the lowest in egg yolks from caged hens. The cholesterol amount was on average 234.4 in one hen egg and 65.2 mg in quail eggs, but in 100 g of eggs, the levels were 385.4 and 545.7 mg, respectively. The levels in 100 g of the edible parts of a chicken egg were 442.3 mg and 634.2 mg in quail eggs.

In the cluster analysis of lipid components, three clusters were distinguished. The first cluster includes most of the examined chicken eggs (E, GL, B, F, C), except for N3. In the second cluster are N3 eggs, and in the third one are quail eggs. The content of lipid compounds in quail eggs significantly differed from their content in chicken eggs. The results are presented in Figure 1. 

The contents of determined fatty acids and cholesterol were subjected to a PCA. As shown by the screen test, four principal components (PCs) were enough to explain 97.3% of the total variance (Table 4).

The highest contribution (about 50%) to the first PC came from the content of C18:0, C20:0, C16:1, C18:3 n6 GLA, C20:2, C20:5 EPA, C22:5, SFA, and lipid quality indices: index of atherogenicity, hypercholesterolemic fatty acids, and the hypocholesterolemic-to-hypercholesterolemic ratio. The levels of C17:0, C:17:1, c11 C18:1, and C18:3 n3 ALA gave their variances to the second PC. In the third PC, the most important value was C18:2 n6 LA, and in the fourth PC, it was the DFA index. A projection of the variables on the factor plane using PCA to represent correlations among lipid components in eggs is shown in Figure 2.

An LDA was used to obtain appropriate classification rules for examined eggs. Relevant discriminant functions were calculated in a stepwise progressive method. In the performed analysis, 19 variables were included in the final model, of which 13 were statistically significant. The applied canonical analysis allowed us to distinguish six discriminant functions, from which five (DF1–DF5) were statistically significant. DF1 was the most significant function, as it explained over 75.15% of discriminatory power. DF2, DF3, DF4, and DF5 explained only 9.61%, 6.18%, 4.70%, and 3.38% of discriminatory power, respectively (Table 5).

The analysis of canonical mean variables indicated that DF1 had the greatest impact on the distinction of quail eggs (Q) from all other groups of eggs. DF2 allowed us to distinguish mostly B eggs from C and N3 eggs. DF3 influenced the distinction of GL eggs from B, C, and N3, and DF4 allowed us to distinguish F eggs mostly from B, GL, and N3, whereas the discriminatory power of DF5 was the weakest regarding E eggs from F and GL. Overall, an excellent separation of quail eggs was obtained.

The calculated classification matrix indicated that the average classification efficiency based on the calculated functions was 71.2% (Table 6). For individual groups, these coefficients were as follows: 100% for Q, 83.3% for GL, 79.2% for B, and 75.0% for E. The lowest coefficients (50.0% and 44.4%) were revealed for F and for N3 eggs, respectively.

The iron content in egg yolks differed between the studied groups. The lowest amount was found in GL eggs (40.7 µg g^−1^), and the highest in the yolk of quail eggs (57.2 µg g^−1^). The amount of Fe in 100 g of chicken eggs was on average 1.26 mg and 2.20 mg in 100 g of quail eggs. For 100 g of edible parts, the results were 1.45 and 2.56, respectively. Zinc levels in egg yolks did not differ significantly between the groups; the mean content of this mineral was 29.1 µg g^−1^. The iron and zinc content in the examined product groups are presented in Table 7.

The analysis of the correlation between lipid components and iron and zinc in chicken and quail eggs showed no clear relationship between them. Although some significant correlations (*p* < 0.05) were found, especially between iron content and some fatty acids, when the results of all groups were analyzed, they were not confirmed in the individual groups of eggs tested. The results are presented in the Appendix A.

## 4. Discussion

Dietary fatty acids affect the level of low- and high-density lipoproteins and thus the dynamic of atherosclerotic plaque formation. They may be pro- or anti-atherogenic and thrombogenic [15]. However, dietary fatty acids affect not only the proper function of the circulatory system in the body but also the immune, neurological, and many other systems [16,17]. Eggs are an important part of the diet for a large proportion of the population, so their fatty acid composition can influence the daily lipid profile of the diet. In 100 g of the edible part of chicken eggs, there is an average of 9.2 g (13.4 g in quail eggs) of fat. The average content of fatty acids in egg fat is 83% [18], which gives 7.7 g of fatty acids (11.1 g for quail) in 100 g of the edible parts of these products.

There are twice as many unsaturated fatty acids as saturated ones in eggs. The dominant FA is oleic acid, which has a beneficial effect on the prevention of CVD [19]. Linoleic acid is also of considerable value as one of the essential unsaturated fatty acids [20]. Unfortunately, the ratio of n6/n3 PUFAs is not the best, so the efforts of producers to increase the content of n3 PUFA seem to be reasonable. The recommended daily intake (RDI) of EPA and DHA in the diet according to various organizations (including the WHO and EFSA) is between 200 and over 600 mg per day, most often around 250 mg [21]. Considering the content of these acids in eggs, the consumption of 100 g of their edible parts (about two chicken eggs) covers over 20% of the RDI, while quail eggs about 37%.

The beneficial composition of fatty acids is best illustrated by the lipid quality indices. These indices were created to approximate the effect of fat on the body [13]. The index of atherogenicity was first described by Ulbricht and Southgate in 1991 [22]. It characterizes the atherogenic potential of FA. Higher values are associated with a greater atherogenic effect on the body. In eggs, this AI is about 0.4. In other animal products, it is usually higher. For example, for fish, it ranges from 0.2 to 1.2 [23,24,25,26]; in red meat, it is 0.3–1.3 [27,28,29,30]; and in milk and its products, it is 1.0–5.0 [31,32,33,34,35]. Naturally, vegetable oils have an incomparably lower AI [36].

The same authors who developed the AI also proposed an index of thrombogenicity [22], referring to the ability to form clots. Like the AI, the higher the TI value, the stronger the thrombogenic effect exerted on the body. The lowest TI among animal products, due to the high share of N3 PUFAs, was found in fish (0.1–0.8) [14,23,26,37]. In the examined eggs, the TI was 0.8–1.0, whereas in red meat, it ranges from 0.8 to 1.6 [28,38,39], and in milk and its products, from 0.4 (yogurt) to 5.0 [33,34,35].

The indices of hypercholesterolemic fatty acids (OFAs) and hypocholesterolemic fatty acids (DFAs) indicate the potential influence on the increase or reduction in the total and LDL cholesterol levels in the blood serum. In 2002, Santos-Silva et al. [40] proposed a hypocholesterolemic-to-hypercholesterolemic ratio. The higher the H/H index, the more beneficial the effect of fat on the body. The H/H ratio for eggs was determined at the level of 2.0–2.3. For fish, it was 0.9–2.9 [24], for red meat, it was 1.2–2.6 [40,41,42], and for dairy products, 0.3–1.3 [33,34,43]. The H/H index for vegetable oils is usually 5.0–15.0 [44,45]. The values of the indices, just like the fatty acid profile (based on which they are calculated), depend on many factors. Nevertheless, egg lipids, compared to other products of animal origin (except for fish oil), are characterized by favorable values of these indicators.

Eggs are a source of high amounts of cholesterol [46]. While there is currently no evidence that dietary cholesterol adversely affects the level of LDL in the body, it is a compound that can be oxidized. Oxidized cholesterol derivatives (COPs) are much better absorbed from the gastrointestinal tract than cholesterol itself [47]. High levels of these compounds in the body are harmful. COPs act in many ways and may contribute to the development of non-communicable diseases [48]. Therefore, there are recommendations to pay attention to the cholesterol level in the diet. Nevertheless, the latest research shows that eating even two eggs a day has no adverse health effects [49,50].

During the evaluation of the nutritional value of eggs, not only lipid component levels but also the amounts of other nutritional elements should be considered. In this study, the content of iron and zinc was also assessed. The daily reference intakes in the European Union for these minerals are 14 and 10 mg, respectively [51]. One hundred grams of the edible part of a hen’s egg covers 10.4% of the RDA for iron (18.3% in quail eggs) and 9.3% for zinc (13.7% in quail eggs). Our results for iron and zinc are consistent with the studies published by other authors [52,53,54].

Consumers perceive organic products as richer in nutrients and healthier. Eggs with a lower number on a shell (denoting the production method) are better received [55]. According to the presented results, there are little (but statistically significant) deviations in the content of tested components between groups. However, based on these results, it is not possible to identify eggs from certain production systems based on the lipid component content or profile (as cluster analysis confirmed). For each tested ingredient in one product group, there was quite a high variation in the results, which could be due to the individual variability, hen age, and different feeds used by different producers. Some other authors have drawn similar conclusions [56,57]. Another issue is the welfare of the animals from which eggs are obtained. Nowadays, animal welfare is a significant concern in a consumer’s decision about animal products, even if it relates to a higher price and the knowledge that there are no significant differences in nutritional value [58,59].

Due to their very high nutritional value, eggs can be considered a functional food [60]. However, producers wanting to meet consumers’ demands and competition modify their products. Eggs enriched with vitamins, minerals, and other bioactive ingredients have been created [10]. In the Polish market, eggs declared to have a high content of omega-3 FAs are quite common. Modification of the egg lipid component profile is usually carried out by modifying the hens’ diet. A series of studies have shown that the addition of ALA-rich linseed to the feed can increase the n3 PUFA content in egg yolks. Alternatively, hens are fed fish oil or microalgae products as a source of long-chain n3 PUFAs. Modifications of the fatty acid profile of eggs were also obtained by feeding hens with vegetable oils or various seeds (chia, hemp) [61,62,63].

In the present study, only products with a clear nutrition claim (high in omega-3 FAs) on the package were chosen from the N3 group. This group differed from the others in its fatty acid profile and had the most favourable values for fat quality indices. However, according to the European Commission Regulation No. 1924/2006 (as amended), “A claim that a food is high in omega-3 fatty acids, and any claim likely to have the same meaning for the consumer, may only be made where the product contains at least 0.6 g α-linolenic acid per 100 g and per 100 kcal or at least 80 mg of the sum of eicosapentaenoic acid and docosahexaenoic acid per 100 g and per 100 kcal” [64]. Unfortunately, none of the eggs in group N3 achieved the level of 0.6 g ALA per 100 g of product or 100 g of the edible parts. Only one producer (out of eight) met the second condition; the sum of EPA and DHA was 95.87 mg 100 g^−1^ of the product (the average in chicken eggs was 48.00 mg 100 g^−1^) and 126.02 mg per 100 g of edible parts (average 55.10 mg in chicken eggs). The seven producers whose eggs were evaluated in this study should not include a nutrition claim on their packages with an increased n3 PUFA content. Many experiments have confirmed the possibility of increasing the n3 PUFA content in eggs [65]. There are also studies of marketed products where the nutrition claim corresponds to an increased amount of these valuable fatty acids [57]. Although the cluster analysis showed that the N3 egg group differed from the others in terms of fatty acid profile, the LDA indicated that it is likely that the dietary changes applied to the laying hens produced multidirectional results that were not necessarily associated with increased n3 PUFA levels. Our research proves that controls on the compatibility of nutrition claims with the real level of the declared nutrients in eggs are necessary.

Another tested premium product is eggs from Green-legged Partridge hens. These eggs have gained popularity in the Polish market due to their better sensory quality than traditional products [66]. Consumers may also believe that they have higher nutritional value. Moreover, their price is higher than other products. The parameters of the egg (the whole egg, edible parts, and yolk weight) were slightly lower than in other groups of chicken eggs, which is a characteristic feature of this breed [67]. The levels of fat, fatty acids, cholesterol, iron, and zinc in the yolk did not differ significantly from other groups of chicken eggs. However, the LDA showed that based on the fatty acid profile, it is possible to distinguish GL eggs from other market eggs with a high probability. Previous studies have presented much more pronounced differences between eggs obtained from different breeds of laying hens [65,68]. But these studies were usually conducted under controlled conditions. In the case of market eggs, many factors affect the final quality of the product, so the differences between breeds can be equalized by other determinants. Only selected parameters of the nutritional quality of eggs were evaluated in this study. Perhaps the differences in the content of other nutrients or bioactive components between eggs from Green-legged Partridge hens and traditional products would be significant.

Quail eggs are quite popular in some regions of the world. Their appearance, size, sensory features, and nutritional value are different from chicken eggs, as they are laid by completely different species of birds [69]. Following this study, attention should be paid to the higher iron and cholesterol contents in quail eggs than in chicken eggs. The differences are even more pronounced when converted into 100 g of whole egg or 100 g of edible parts. The share of yolk in a quail egg is about 10% higher than that in a hen’s egg (38.5 vs. 28.1%). Although the percentage share of the main groups of FAs (SFAs, MUFAs, PUFAs) in the total FA content is similar, the content of individual acids distinguishes quail eggs from the other studied groups. The data for quail eggs presented in this study are consistent with the results of studies obtained by other authors [69,70,71]. Quail eggs seem to be an interesting alternative to chicken eggs in the kitchen.

## 5. Conclusions

Bird eggs are an almost perfect product, intended to provide the bird embryo with all the nutrients necessary for development. Therefore, they also play a significant role in people’s diets. Compared to other products of animal origin, the fatty acid profile is favorable, and modification of the laying hens’ diet may contribute to an increase in the level of n3 fatty acids, especially valuable EPA and DHA. Tested eggs that were labeled as high in n-3 PUFAs (N3) had a slightly different fatty acid profile to regular ones. However, eggs from only one producer achieved the levels of EPA and DHA required to be able to use a nutritional claim about the higher content of this FA on the packaging. In this study, there were also no differences in the content of tested nutrients between eggs from different production methods (E, F, B, C). Although GL eggs did not differ significantly from other hen eggs in the content of individual lipid components, an LDA can be used to distinguish eggs from different laying breeds with a high degree of confidence based on the fatty acid profile. Quail eggs differed in terms of FA profiles, as well as iron and cholesterol levels, from chicken eggs. Meals prepared from them can be an interesting alternative to a traditional omelet.

## Figures and Tables

**Figure 1 foods-13-01571-f001:**
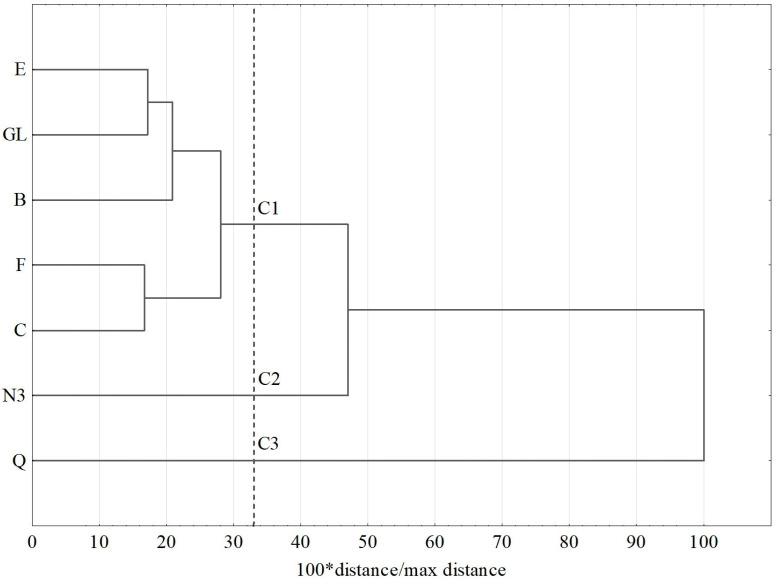
Dendrogram of similarity in fatty acid and cholesterol contents in yolk fat of investigated groups of eggs; C1–C3—clusters; E—chicken eggs from organic production; F—free-range chicken eggs; B—barn chicken eggs; C—chicken eggs from caged hens; N3—chicken eggs with an increased content of n3 fatty acids; GL—chicken eggs from Green-legged Partridge; Q—partridge quail eggs.

**Figure 2 foods-13-01571-f002:**
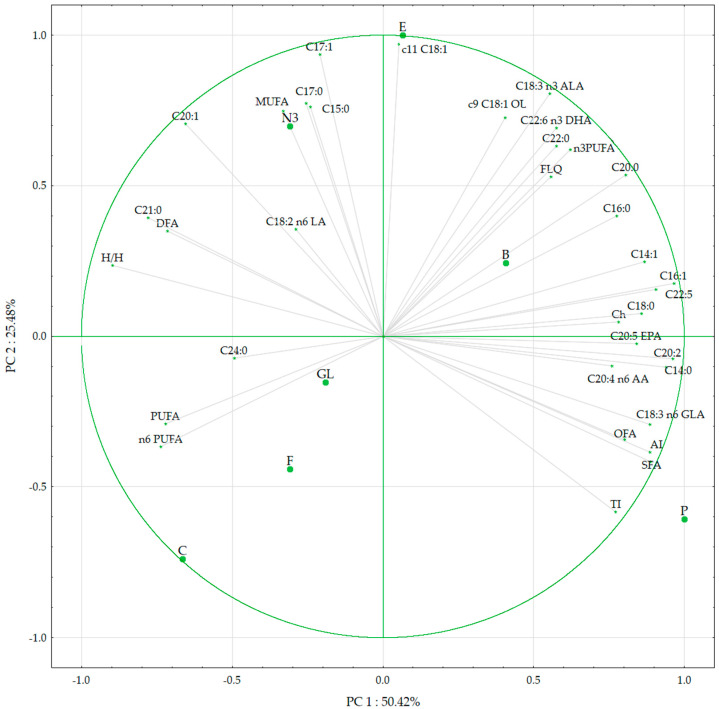
Projection of the variables on the factor plane using principal component analysis to present correlations among lipid components in chicken and quail eggs; biplot between PC 1 and PC 2 showing the divided groups. E—chicken eggs from organic production; F—free-range chicken eggs; B—barn chicken eggs; C—chicken eggs from caged hens; N3—chicken eggs with an increased content of n3 fatty acids; GL—chicken eggs from Green-legged Partridge; Q—partridge quail eggs; Ch—cholesterol; FLQ—flesh-lipid quality; AI—index of atherogenicity; TI—index of thrombogenicity; OFAs—hypercholesterolemic fatty acids; DFAs—hypocholesterolemic fatty acids; H/H—hypocholesterolemic-to-hypercholesterolemic ratio.

**Table 1 foods-13-01571-t001:** Characteristics of examined egg samples ($\bar{x}$ ± SD).

Group	E	F	B	C	N3	GL	Q	*p*-Value
Egg [g]	60.0 ± 4.7 ^b^	62.9 ± 2.0 ^a,b^	62.0 ± 4.8 ^a,b^	61.3 ± 3.6 ^a,b^	64.2 ± 3.9 ^a^	56.4 ± 4.5	11.9 ± 1.0	<0.001
Yolk [g]	17.0 ± 1.8 ^a^	17.1 ± 1.1 ^a^	17.0 ± 1.7 ^a^	17.1 ± 1.3 ^a^	17.4 ± 1.7 ^a^	16.9 ± 2.0 ^a^	4.60 ± 0.69	<0.001
Eggshell [g]	7.46 ± 0.56 ^b,c^	7.95 ± 0.50 ^a,b^	8.24 ± 1.14 ^a,b^	8.02 ± 0.93 ^a,b^	8.37 ± 0.87 ^a^	7.12 ± 0.63 ^c^	1.66 ± 0.18	<0.001
Edible part [g]	52.5 ± 4.3 ^a^	55.0 ± 1.9 ^a^	53.7 ± 4.4 ^a^	53.3 ± 3.2 ^a^	55.8 ± 3.4 ^a^	49.3 ± 4.1	10.2 ± 0.9	<0.001
Yolk fat [%]	32.0 ± 7.1 ^a^	29.5 ± 2.5 ^a,b^	29.6 ± 4.8 ^a,b^	25.7 ± 5.0 ^b^	28.7 ± 7.2 ^a,b^	26.9 ± 4.2 ^b^	30.0 ± 5.1 ^a,b^	0.002

E—chicken eggs from organic production; F—free-range chicken eggs; B—barn chicken eggs; C—chicken eggs from caged hens; N3—chicken eggs with an increased content of n3 fatty acids; GL—chicken eggs from Green-legged Partridge; Q—partridge quail eggs. *p*-value; result of one-way ANOVA (α = 0.05). ^a–c^ homogeneous groups in rows; comparison between product groups (Tukey’s test, α = 0.05).

**Table 2 foods-13-01571-t002:** Fatty acid and cholesterol contents in examined egg yolks ($\bar{x}$ ± SD).

Group	E	F	B	C	N3	GL	Q	*p*-Value
Fatty acids	[mg g^−1^]							
C14:0	0.96 ± 0.26 ^a^	0.86 ± 0.15 ^a,b^	0.92 ± 0.19 ^a,b^	0.74 ± 0.20 ^b^	0.81 ± 0.22 ^a,b^	0.82 ± 0.23 ^a,b^	1.25 ± 0.26	<0.001
C15:0	0.26 ± 0.14 ^a,b^	0.22 ± 0.05 ^a,b^	0.28 ± 0.14 ^a,b^	0.19 ± 0.07 ^b^	0.39 ± 0.44 ^a^	0.24 ± 0.13 ^a,b^	0.18 ± 0.05 ^b^	0.011
C16:0	65.3 ± 15.7 ^a^	61.2 ± 5.8 ^a,b^	64.2 ± 10.5 ^a^	52.7 ± 11.4 ^b^	58.2 ± 15.8 ^a,b^	56.0 ± 10.0 ^a,b^	64.6 ± 12.1 ^a^	0.002
C17:0	0.59 ± 0.17 ^a^	0.53 ± 0.08 ^a,b^	0.51 ± 0.18 ^a,b^	0.47 ± 0.16 ^b^	0.52 ± 0.18 ^a,b^	0.49 ± 0.11 ^a,b^	0.44 ± 0.05 ^b^	0.031
C18:0	20.9 ± 4.8 ^a,b^	19.2 ± 2.4 ^b,c^	18.7 ± 3.4 ^b,c^	15.5 ± 2.6 ^d^	17.0 ± 4.4 ^c,d^	17.7 ± 3.0 ^c,d^	22.8 ± 3.8 ^a^	<0.001
C20:0	0.13 ± 0.04 ^a^	0.12 ± 0.03 ^a,b^	0.14 ± 0.03 ^a^	0.10 ± 0.04 ^b^	0.13 ± 0.05 ^a,b^	0.12 ± 0.03 ^a,b^	0.14 ± 0.04 ^a^	0.002
C21:0	0.36 ± 0.13 ^a^	0.36 ± 0.10 ^a^	0.31 ± 0.11 ^a^	0.32 ± 0.12 ^a^	0.32 ± 0.18 ^a^	0.34 ± 0.14 ^a^	0.17 ± 0.05	<0.001
C24:0	0.56 ± 0.22 ^b,c^	0.90 ± 0.44 ^a^	0.79 ± 0.37 ^a,b^	0.72 ± 0.38 ^a,b^	0.67 ± 0.58 ^a,b^	0.52 ± 0.24 ^b,c^	0.43 ± 0.20 ^c^	<0.001
SFA	89.1 ± 20.6 ^a^	83.4 ± 8.0 ^a,b^	85.9 ± 14.2 ^a^	70.9 ± 14.0 ^b^	78.1 ± 21.5 ^a,b^	76.2 ± 13.2 ^a,b^	90.1 ± 15.9 ^a^	<0.001
C14:1	0.20 ± 0.11 ^a,b^	0.17 ± 0.06 ^a,b^	0.23 ± 0.08 ^a^	0.14 ± 0.05 ^b^	0.17 ± 0.09 ^a,b^	0.19 ± 0.13 ^a,b^	0.22 ± 0.05 ^a,b^	0.008
C16:1	8.65 ± 3.71 ^a–c^	7.64 ± 1.93 ^c^	10.04 ± 2.23 ^a,b^	6.62 ± 2.09 ^c^	8.28 ± 2.89 ^a–c^	7.90 ± 2.77 ^b,c^	10.59 ± 2.96 ^a^	<0.001
C17:1	0.42 ± 0.11 ^a^	0.33 ± 0.06 ^b,c^	0.39 ± 0.12 ^a,b^	0.31 ± 0.09 ^c^	0.40 ± 0.13 ^a,b^	0.35 ± 0.05 ^a–c^	0.28 ± 0.06 ^c^	<0.001
c9 C18:1 OL	106.3 ± 24.0 ^a^	96.3 ± 10.6 ^a,b^	95.4 ± 15.7 ^a,b^	85.0 ± 16.3 ^b^	96.3 ± 22.2 ^a,b^	89.5 ± 11.5 ^b^	96.1 ± 20.6 ^a,b^	0.004
c11 C18:1	8.95 ± 3.30 ^a^	7.32 ± 1.30 ^a–c^	8.25 ± 1.94 ^a,b^	6.57 ± 1.26 ^b,c^	7.55 ± 2.03 ^a,b^	7.45 ± 1.77 ^a,b^	5.35 ± 1.09 ^c^	<0.001
C20:1	0.48 ± 0.15 ^a^	0.41 ± 0.06 ^a^	0.41 ± 0.11 ^a^	0.39 ± 0.10 ^a^	0.50 ± 0.20 ^a^	0.46 ± 0.12 ^a^	0.25 ± 0.09	<0.001
MUFA	124.2 ± 30.0 ^a^	110.1 ± 13.4 ^a,b^	113.7 ± 18.1 ^a,b^	97.4 ± 19.2 ^b^	112.8 ± 25.8 ^a,b^	104.7 ± 15.1 ^b^	112.5 ± 24.5 ^a,b^	0.002
C18:2 LA	42.5 ± 12.5	42.8 ± 9.2	36.7 ± 10.4	37.6 ± 10.4	38.5 ± 14.2	34.6 ± 9.0	35.9 ± 4.6	0.050
C18:3 GLA	0.28 ± 0.09 ^a^	0.27 ± 0.07 ^a^	0.29 ± 0.10 ^a^	0.22 ± 0.08 ^a^	0.26 ± 0.10 ^a^	0.25 ± 0.08 ^a^	0.53 ± 0.16	<0.001
C18:3 ALA	2.12 ± 0.61 ^a^	1.38 ± 0.81 ^b^	2.00 ± 1.71 ^a,b^	1.34 ± 0.86 ^b^	1.85 ± 0.93 ^a,b^	1.55 ± 0.49 ^a,b^	1.78 ± 1.08 ^a,b^	0.047
C20:2	0.09 ± 0.05 ^a^	0.07 ± 0.02 ^a^	0.09 ± 0.03 ^a^	0.06 ± 0.03 ^a^	0.08 ± 0.05 ^a^	0.09 ± 0.04 ^a^	0.15 ± 0.07	<0.001
C20:3	0.32 ± 0.08	0.27 ± 0.05	0.29 ± 0.07	0.26 ± 0.09	0.27 ± 0.1	0.30 ± 0.07	0.29 ± 0.09	0.180
C20:4 AA	4.54 ± 1.16 ^a,b^	4.54 ± 0.81 ^a,b^	4.12 ± 0.96 ^a–c^	3.68 ± 0.91 ^c^	3.78 ± 1.12 ^b,c^	3.93 ± 0.67 ^b,c^	5.01 ± 1.16 ^a^	<0.001
C20:5 EPA	0.06 ± 0.06 ^a,b^	0.03 ± 0.01 ^b^	0.09 ± 0.07 ^a,b^	0.03 ± 0.01 ^b^	0.06 ± 0.05 ^a,b^	0.03 ± 0.02 ^b^	0.11 ± 0.07 ^a^	0.001
C22:2	0.13 ± 0.09	0.14 ± 0.08	0.14 ± 0.13	0.17 ± 0.13	0.12 ± 0.08	0.18 ± 0.26	0.11 ± 0.05	0.622
C22:5	0.29 ± 0.10 ^a,b^	0.18 ± 0.08 ^c^	0.24 ± 0.13 ^b,c^	0.17 ± 0.10 ^c^	0.21 ± 0.11 ^b,c^	0.19 ± 0.10 ^c^	0.36 ± 0.06 ^a^	<0.001
C22:6 DHA	2.12 ± 0.46 ^a^	1.37 ± 0.61 ^b^	1.81 ± 0.74 ^a,b^	1.45 ± 0.81 ^b^	1.96 ± 1.10 ^a,b^	1.49 ± 0.27 ^b^	1.97 ± 0.29 ^a,b^	<0.001
PUFA	52.4 ± 14.0	51.1 ± 9.2	45.8 ± 12.5	44.9 ± 12.2	47.0 ± 15.7	42.6 ± 10.3	46.2 ± 5.2	0.062
n3	4.28 ± 1.01 ^a^	2.76 ± 1.38 ^a,b^	3.84 ± 2.41 ^a,b^	2.80 ± 1.64 ^b^	3.84 ± 2.01 ^a,b^	3.06 ± 0.67 ^a,b^	3.85 ± 1.33 ^a,b^	0.004
n6	48.1 ± 13.6	48.3 ± 9.5	41.9 ± 11.3	42.1 ± 11.4	43.2 ± 15.4	39.5 ± 9.8	42.3 ± 5.6	0.063
Cholesterol	[mg g^−1^]							
	13.9 ± 0.6 ^a–c^	14.1 ± 0.8 ^a–c^	14.4 ± 0.6 ^a^	13.1 ± 0.6 ^d^	13.3 ± 0.9 ^c,d^	13.6 ± 0.9 ^b–d^	14.2 ± 1.3 ^a,b^	<0.001

E—chicken eggs from organic production; F—free-range chicken eggs; B—barn chicken eggs; C—chicken eggs from caged hens; N3—chicken eggs with an increased content of n3 fatty acids; GL—chicken eggs from Green-legged Partridge; Q—partridge quail eggs. *p*-value, result of one-way ANOVA (α = 0.05). ^a–d^ homogeneous groups in rows; comparison between product groups (Tukey’s test, α = 0.05).

**Table 3 foods-13-01571-t003:** The share of the main groups of fatty acids in the total FA content and fat quality indices ($\bar{x}$ ± SD) in egg yolks.

Group	E	F	B	C	N3	GL	Q	*p*-Value
SFA [%]	33.5 ± 1.18 ^c^	34.1 ± 1.0 ^b,c^	35.0 ± 0.89 ^a,b^	33.3 ± 1.2 ^c^	32.8 ± 2.2 ^c^	34.1 ± 1.5 ^b,c^	36.2 ± 1.5 ^a^	<0.001
MUFA [%]	46.7 ± 3.8	45.0 ± 3.8	46.4 ± 3.3	45.7 ± 3.8	47.6 ± 3.2	47.0 ± 2.4	44.9 ± 3.2	0.100 *
PUFA [%]	19.8 ± 3.9	20.9 ± 3.7	18.6 ± 3.48	21.0 ± 3.40	19.6 ± 3.13	19.9 ± 2.8	18.9 ± 2.7	0.086 *
n3 PUFA [%]	1.64 ± 0.37 ^a^	1.13 ± 0.56 ^b^	1.57 ± 1.02 ^a^	1.26 ± 0.56 ^b^	1.65 ± 0.84 ^a^	1.37 ± 0.23 ^a,b^	1.66 ± 0.90 ^a^	0.042
n6 PUFA [%]	18.2 ± 3.8 ^a,b^	19.8 ± 3.9 ^a^	17.0 ± 2.9 ^b^	19.7 ± 3.6 ^a^	17.9 ± 2.9 ^a,b^	17.6 ± 2.7 ^a,b^	17.2 ± 2.1 ^a,b^	0.015
n6/n3 PUFA	10.7 ± 3.0 ^c^	24.2 ± 18.5 ^a^	12.5 ± 4.7 ^b,c^	20.3 ± 14.6 ^a,b^	13.4 ± 8.1 ^b,c^	12.1 ± 1.8 ^b,c^	10.6 ± 2.8 ^c^	<0.001
FLQ	0.83 ± 0.17 ^a^	0.57 ± 0.25 ^b^	0.75 ± 0.30 ^a,b^	0.66 ± 029 ^a,b^	0.86 ± 0.47 ^a^	0.69 ± 0.14 ^a,b^	0.86 ± 0.26 ^a^	0.003
AI	0.39 ± 0.02 ^b^	0.40 ± 0.02 ^b^	0.43 ± 0.02 ^a^	0.39 ± 0.03 ^b^	0.39 ± 0.04 ^b^	0.40 ± 0.03 ^b^	0.44 ± 0.03 ^a^	<0.001
TI	0.88 ± 0.06 ^c,d^	0.93 ± 0.05 ^a–c^	0.94 ± 0.07 ^a,b^	0.89 ± 0.07 ^b–d^	0.85 ± 0.12 ^d^	0.91 ± 0.07 ^b–d^	0.99 ± 0,09 ^a^	<0.001
OFA	24.9 ± 1.1 ^c^	25.4 ± 1.0 ^b,c^	26.6 ± 1.0 ^a^	25.0 ± 1.2 ^c^	24.8 ± 1.7 ^c^	25.7 ± 1.5 ^b,c^	26.4 ± 0.9 ^a,b^	<0.001
DFA	74.6 ± 1.1 ^a^	73.7 ± 0.9 ^a,b^	72.6 ± 1.0 ^c^	74.1 ± 1.3 ^a,b^	74.4 ± 1.8 ^a^	73.9 ± 1.5 ^a,b^	73.0 ± 1.5 ^b,c^	<0.001
H/H	2.29 ± 0.19 ^a^	2.27 ± 0.17 ^a^	2.06 ± 0.14 ^c^	2.33 ± 0.17 ^a^	2.33 ± 0.27 ^a^	2.23 ± 0.19 ^a,b^	2.04 ± 0.14 ^b,c^	<0.001

FLQ—flesh-lipid quality; AI—index of atherogenicity; TI—index of thrombogenicity; OFAs—hypercholesterolemic fatty acids; DFAs—hypocholesterolemic fatty acids; H/H—hypocholesterolemic-to-hypercholesterolemic ratio; E—chicken eggs from organic production; F—free-range chicken eggs; B—barn chicken eggs; C—chicken eggs from caged hens; N3—chicken eggs with an increased content of n3 fatty acids; GL—chicken eggs from Green-legged Partridge; Q—partridge quail eggs. *p*-value, result of one-way ANOVA (α = 0.05); * lack of differences between product groups. ^a–d^ homogeneous groups in rows; comparison between product groups (Tukey’s test, α = 0.05).

**Table 4 foods-13-01571-t004:** Loadings, eigenvalues, and variances of the significant principal components (PCs) of chicken and quail egg yolk lipids.

Variables	PC 1	PC 2	PC 3	PC 4
C16:0	0.77	0.42	−0.47	−0.03
C17:0	−0.26	0.81	−0.50	−0.07
C18:0	0.85	0.10	−0.34	−0.34
C20:0	0.81	0.52	−0.11	0.23
C21:0	−0.78	0.41	−0.36	0.22
C22:0	0.57	0.64	−0.44	0.23
C24:0	−0.50	−0.07	−0.60	0.41
C16:1	0.97	0.16	0.02	0.15
C17:1	−0.20	0.93	−0.07	0.29
c9 C18:1 n9 OL	0.41	0.75	−0.43	−0.28
c11 C18:1	0.06	0.97	0.05	0.20
C20:1	−0.65	0.70	0.06	0.20
C18:2 n6 LA	−0.29	0.40	−0.76	−0.39
C18:3 n6 GLA	0.89	−0.29	0.04	−0.34
C18:3 n3 ALA	0.56	0.80	0.08	0.07
C20:2	0.96	−0.08	0.08	−0.18
C20:4 n6 AA	0.75	−0.06	−0.50	−0.39
C20:5 EPA	0.85	−0.03	0.27	−0.45
C22:5	0.91	0.16	0.10	−0.35
C22:6 n3 DHA	0.59	0.69	0.22	−0.28
SFA	0.88	−0.42	−0.14	0.13
MUFA	−0.32	0.71	0.54	0.28
PUFA	−0.73	−0.25	−0.40	−0.45
n3PUFA	0.63	0.60	0.45	−0.11
n6 PUFA	−0.74	−0.33	−0.43	−0.37
AI	0.88	−0.40	−0.07	0.24
TI	0.76	−0.58	−0.22	0.13
FLQ	0.57	0.51	0.53	−0.30
OFA	0.80	−0.36	−0.10	0.45
DFA	−0.71	0.37	0.16	−0.55
H/H	−0.90	0.26	0.03	−0.36
Cholesterol	0.77	0.06	−0.56	0.28
Eigenvalue	31.1			
Variance (%)	49.9	26.0	12.4	9.0
Cumulative (%)	97.3			

FLQ—flesh-lipid quality; AI—index of atherogenicity; TI—index of thrombogenicity; OFAs—hypercholesterolemic fatty acids; DFAs—hypocholesterolemic fatty acids; H/H—hypocholesterolemic-to-hypercholesterolemic ratio.

**Table 5 foods-13-01571-t005:** Coefficients and average value of canonical variables included in the final LDA model of lipid components in chicken and quail eggs.

	Coefficients of Canonical Variables					
	DF1	DF2	DF3	DF4	DF5	DF6
Discriminatory power	75.15%	9.61%	6.18%	4.70%	3.38%	0.97%
Variables						
C18:3 n6 GLA	1.31	−0.65	−0.25	−1.05	0.49	0.67
C21:0	−0.56	0.16	0.24	−0.59	0.37	−1.06
C17:1	−0.45	0.65	0.05	−0.46	−0.71	0.21
C24:0	−0.80	0.31	−0.36	0.53	−0.17	0.32
C18:0	1.52	0.67	0.39	−0.39	−0.72	−0.16
C18:2 n6 LA	0.55	−0.91	−2.34	1.23	−1.50	0.54
Cholesterol	0.16	0.61	−0.01	0.14	−0.04	0.25
c11 C20:1	−0.35	−0.68	0.63	−0.33	0.02	0.34
C17:0	−0.63	0.34	0.57	−0.34	0.57	0.01
C20:4 n6 AA	0.16	−0.31	1.67	0.80	0.81	0.03
C16:0	−1.74	2.23	0.11	0.20	1.66	−1.07
c9 C18:1 OL	−0.28	−1.29	−0.32	0.71	−0.71	1.04
C22:6 n3 DHA	0.26	−0.04	−2.33	−0.13	−1.22	−1.88
C18:3 n3 ALA	−0.75	0.37	2.25	0.37	0.83	1.34
C16:1	1.50	−0.24	−1.01	−0.83	−0.16	0.92
C15:0	0.11	−0.41	−0.12	−0.07	−0.19	0.35
C14:0	0.68	−0.33	0.71	0.36	−0.51	−0.52
C14:1	−0.85	−0.041	−0.09	0.13	−0.13	−0.44
C20:5 EPA	0.20	−0.35	0.04	0.30	−0.26	0.10
	Average values of canonical variables					
E	−0.62	0.39	0.56	0.45	−0.92	−0.21
F	−0.64	0.28	0.17	0.96	0.45	0.37
B	−0.32	1.60	−0.69	−0.53	0.15	−0.05
C	−1.47	−0.99	−0.69	0.28	0.35	−0.38
N3	−1.04	−1.02	−0.63	−0.60	−0.62	0.44
GL	−0.68	−0.30	1.19	−0.74	0.42	−0.03
Q	7.20	−0.34	−0.10	0.04	0.019	−0.03

E—chicken eggs from organic production; F—free-range chicken eggs; B—barn chicken eggs; C—chicken eggs from caged hens; N3—chicken eggs with an increased content of n3 fatty acids; GL—chicken eggs from Green-legged Partridge; Q—partridge quail eggs

**Table 6 foods-13-01571-t006:** Results of the LDA presenting the correct classification percentage and the predicted group membership for actual groups of examined eggs.

		Predicted Group Membership						
Actual Group	Correct Classification	E	F	B	C	N3	GL	Q
E	75.0%	18	1	2	2	1	0	0
F	50.0%	6	12	3	1	0	2	0
B	79.2%	1	2	19	1	1	0	0
C	70.8%	0	0	1	17	2	4	0
N3	44.4%	2	1	2	3	8	2	0
GL	83.3%	0	1	2	1	0	20	0
Q	100.0%	0	0	0	0	0	0	15
Σ	71.2%	27	17	29	25	12	28	15

E—chicken eggs from organic production; F—free-range chicken eggs; B—barn chicken eggs; C—chicken eggs from caged hens; N3—chicken eggs with an increased content of n3 fatty acids; GL—chicken eggs from Green-legged Partridge; Q—partridge quail eggs.

**Table 7 foods-13-01571-t007:** Iron and zinc content in egg yolks ($\bar{x}$ ± SD).

Group	E	F	B	C	N3	GL	Q	*p*-Value
Iron [µg g^−1^]	43.5 ± 7.0 ^a,b^	49.1 ± 8.8 ^a^	49.3 ± 6.6 ^a^	44.1 ± 6.8 ^a,b^	47.7 ± 5.9 ^a,b^	40.7 ± 8.0 ^b^	57.2 ± 6.8	<0.001
Zinc [µg g^−1^]	28.6 ± 3.3	29.3 ± 3.1	28.2 ± 2.0	28.5 ± 3.2	29.8 ± 2.4	29.4 ± 3.1	30.5 ± 4.0	0.216

E—chicken eggs from organic production; F—free-range chicken eggs; B—barn chicken eggs; C—chicken eggs from caged hens; N3—chicken eggs with an increased content of n3 fatty acids; GL—chicken eggs from Green-legged Partridge; Q—partridge quail eggs. *p*-value, result of one-way ANOVA (α = 0.05). ^a,b^ homogeneous groups in a row; comparison between product groups (Tukey’s test, α = 0.05).

## Data Availability

The original contributions presented in the study are included in the article/Appendix A, further inquiries can be directed to the corresponding author.

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
