# Peer review of "Evaluation and Discrimination of Lipid Components and Iron and Zinc Levels in Chicken and Quail Eggs Available on the Polish Market"

_foods, 2024, doi:10.3390/foods13101571_

Round 1
Reviewer 1 Report
Comments and Suggestions for Authors
Review on the manuscript “Comparison of Lipid Components, Iron and Zinc Levels in 2 Chicken and Quail Eggs Available on the Market”
This work determines the lipid components and content of iron and zinc in the chicken and quail eggs. The results obtained in this article show that bird eggs are a source of valuable nutrients for humans and will have high health benefits. Complete studies have been made on the fatty acid composition of eggs yolk and on content cholesterol as well as content of iron and zinc. In addition, a cluster analysis of lipid components content was performed. The research materials are described in Materials and methods, as well as the analytical methods of analysis. Modern methods have been used to analyze eggs yolk. The chosen research methods allow achieving the set aim in the manuscript. The tables and figure are clear and they accurately represent the results. The results are interpreted well. In the discussion the authors they compare the results obtained in this work with those of other authors. The results support the conclusions. The abstract accurately reflects the results of the study. There are 71 sources cited, 69 of which are after the year 2000 and 39 are after the year 2015. The authors use relatively new literature sources.
I have some technical remarks on the article.
In the abstract line 15 it written “un-saturated acids”, but should be “unsaturated acids”.
In Introduction line 53 and 56 – It is written “bioac-tive ingredients and how-ever”, but should be “bioactive ingredients and however”.
In Material and methods in section 2.2.2. Fatty acids analysis
It is written “Then 1 mL of BF3 solution in methanol” and “Chromatographic separations were conducted on a capillary column SGE BPX70 (60 m / 0,25 mm ID / film thickness 0,20 μm; Ringwood, Australia). Helium was used as the carrier gas (flow: linear velocity at 0.9 mL min-1)”
Should be “Then 1 mL of BF3 solution in methanol” and “Chromatographic separations were conducted on a capillary column SGE BPX70 (60 m / 0.25 mm ID / film thickness 0.20 μm; Ringwood, Australia). Helium was used as the carrier gas (flow: linear velocity at 0.9 mL min-1)”.
In Results – line 231 It is written “Cholesterol levels ranged from 13.09 mg g-1 to 14.35 mg g-1.”, but should be ““Cholesterol levels ranged from 13.1 mg g-1 to 14.4 mg g-1.” These values (13.1 and 14.4) are given in Table 2.
In Discussion – line 349 and 350 - It is written “95.87 mg 100g-1 of the product (average in chicken 349 eggs was 48.00 mg 100 g-1)”, should be “95.87 mg 100g-1 of the product (average in chicken 349 eggs was 48.00 mg 100 g-1)”.
Line 366 and 367 – However, the word is repeated. Both sentences begin with the word however.
My comments are given in the attached file.

Author Response
Dear Reviewer,
We are very grateful for your time and valuable comments.
All suggested corrections have been incorporated into the article. Most of them are the result of retyping the text from the draft version into the template recommended by Foods. We have also reread the article and corrected minor punctuation and stylistic errors.
Thank you for your help in improving our article.

Reviewer 2 Report
Comments and Suggestions for Authors
The authors compared the content of lipid components and their nutritional value as well as iron and zinc levels in chicken and quail eggs. Specific comments are shown as following:
1. In subsection 2.2.2., formula for calculating lipid quality indicators can be further explained. Please provide more details on de-aromatization techniques.
2. In line 358, From the results, it seems that eggs from Green-legged partridge hens are not better than other group eggs.
3. In conclusions. “modification of the …valuable EPA and DHA”. Please provide more details.
Author Response
Dear Reviewer,
We are very grateful for your time and valuable comments. The specific comments made in the Review are addressed below.
Thank you for your help in improving our article.
R: In subsection 2.2.2., the formula for calculating lipid quality indicators can be further explained. Please provide more details on de-aromatization techniques.
A: Thank you very much for this comment. We have added an explanation of all the symbols directly below the formulas.
The second part of the question, I think, was about mineralisation. This process was carried out in a very simple (but foolproof) device, a microwave mineraliser, where it was only possible to mineralise one sample at a time. On this device, we set the time, temperature, power, and maximum pressure. All these parameters are given in the article.
R: In line 358, From the results, it seems that eggs from Green-legged partridge hens are not better than other group eggs.
A: In Poland, eggs from green-legged hens are considered a premium product over regular hen eggs. They are considered to have better organoleptic properties and some (as we stated in the quoted sentence) consider them to have better nutritional value. They have a slightly higher price. Our research has shown that they do not differ from normal eggs in terms of their fatty compound content and profile, or their iron and zinc content. However, we cannot exclude the possibility that there are differences in the nutritional value of eggs from different breeds of laying hens (i.e. in the content of vitamins, minerals or some bioactive substances). Our study was limited to only one part of the nutritional value, so we have to be cautious when making statements.
R: In conclusions. “modification of the …valuable EPA and DHA”. Please provide more details.
A: In the conclusions we have added: “Tested eggs, that were labelled as high in n-3 PUFA (N3), had a slightly different fatty acid profile to regular ones. However, eggs from only one producer achieved the levels of EPA and DHA to be able to use nutritional claim about higher content of this FA on the packaging.”
Reviewer 3 Report
Comments and Suggestions for Authors
The manuscript tried to investigate the impact of different commercial chicken and quail eggs available on the local market on the body physiology. These results may provide a potential approach in the potential impacts of the phytochemical molecules for the human health. On the other hand, the study design was not well prepared. I suggest that the authors should work more to present that study in the best way to be considered in Foods journal. I think the experiment integration was not well explained and appeared to be rather convoluted. A lot of statements and conclusions presented were quite speculative like what does the authors mean by “body physiology”? The authors just focused on few in vitro chemical measurements that related to the eggs from the different franchises. without any significant contribution to the physiological impact that related to human body or even the animal in vivo studies. Thus, the current manuscript version could be revisied.
Generally, the title and the abstract sections should be more focused on your study objectives with the used phytochemical name. In which the results and findings must be more clarified in the abstract. For instance, the authors did not mention the novel results and findings from the study which related to the experiments. Like, serving size relationship to the market and full information about the sources with showing flowchart figure for their differences where their significant contributions could not be concluded from the author study. The number of used types, 6 types, is not enough for comprehensive presentation of the exact expression of the overall population.
In the materials and methods: What do you mean with “The content of saturated fatty acids (SFA) ranged from 32.8% (N3) to 36.2% (Q) of total fatty acids (TFA).”? Why the authors just focused on the mentioned two elements? What will be the message the authors want to deliver from the overall discrimination? Several concerns are raised against integrating the different systems applicability like in case of your mentioned fish sources in your discussion. I think the overall analysis are so simple and needs more efforts to have the conclusions that your Bird eggs samples are almost perfect sources of all the nutrients necessary for development. Even, development of what? Also, have the authors made a positive and negative controls or just chemical discrimination by chromatographic analysis? Where the physiological measurements that should clarify the authors’ points. The presentation of the data are so simple and no integrations among the presented data.
Comments on the Quality of English Language
Moderate editing of English language required.
Author Response
Dear Reviewer,
Thank you for taking the time to review our article.
However, we cannot respond positively to most of the comments. We have provided detailed responses to each of the objections to our article below.
R: The manuscript tried to investigate the impact of different commercial chicken and quail eggs available on the local market on body physiology.
A: We would like to emphasize that this is not the main objective of our studies. As stated in lines 58-60 “The study aimed to compare the content of lipid components and their nutritional value as well as iron and zinc levels in chicken and quail eggs commonly available on the market.” We did not investigate the impact of study material on body physiology. There is no experiment regarding this topic described in our manuscript. We would like to express our great astonishment and concern regarding this remark of Reviewer 3 and we would like to ask about the reasons for such a statement.
R: These results may provide a potential approach in the potential impacts of the phytochemical molecules for the human health.
A: We would like to emphasize that neither phytochemicals nor their impact on human health were investigated in this study and described in this manuscript. We have this overwhelming impression that this statement does not refer to our manuscript, similar to the previous statement of Reviewer 3. One more time we would like to ask about the reasons for such a statement.
R: On the other hand, the study design was not well prepared. I suggest that the authors should work more to present that study in the best way to be considered in Foods journal. I think the experiment integration was not well explained and appeared to be rather convoluted.
A: Regarding aims and scopes of Foods (https://www.mdpi.com/journal/foods/about) we would like to mention the first sentence: “…advanced forum for studies related to all aspects of food research, with major emphasis on the “science of food””. In our opinion studies presenting the comparison of nutrients in different types of food, especially food products available on the market, are very important and meet the criteria of food research. Moreover, as explained in the ‘introduction’ section our study investigated all types of hen eggs available in the European Union (0, free-range eggs have number 1, barn eggs 2, and cage eggs 3) as well as eggs enriched in n3 fatty acids, eggs from Green-legged partridge and eggs from partridge quail. It gives the overview of the vast majority of types of eggs available for the regular consumers in European Union. We strongly disagree with the statement as experiment integration was not well explained and was rather convoluted. We would like to ask for withdraw of this statement or we would like to ask Reviewer for the explanation of their opinion based on the manuscript text.
R: A lot of statements and conclusions presented were quite speculative like what does the authors mean by “body physiology”?
A: We would like to express one more time our astonishment and concern as in our manuscript there is no such expression as “body physiology” used. We kindly ask Reviewer 3 for the example of such statement in our text.
R: The authors just focused on few in vitro chemical measurements that related to the eggs from the different franchises. without any significant contribution to the physiological impact that related to human body or even the animal in vivo studies. Thus, the current manuscript version could be revisied.
A: We appreciate this remark of Reviewer 3 but we would like explain one more time that investigation of the impact of examined eggs on physiological process of human body or performing of animal model studies was not the aim of our investigation. We would like to ensure that we will take this hint into consideration in planning of our future research.
R: Generally, the title and the abstract sections should be more focused on your study objectives with the used phytochemical name.
A: As mentioned before no phytochemicals were used or investigated in our study. We kindly ask Reviewer 3 for the strict justification of this statement based on the manuscript text or, preferably, we would like to ask the for withdrawal of this remark as it is not related to our study.
R: In which the results and findings must be more clarified in the abstract. For instance, the authors did not mention the novel results and findings from the study which related to the experiments. Like, serving size relationship to the market and full information about the sources with showing flowchart figure for their differences where their significant contributions could not be concluded from the author study.
The number of used types, 6 types, is not enough for comprehensive presentation of the exact expression of the overall population.
A: As explained in the ‘introduction’ our study covered all types of hen eggs available on the marked in European Union as well as well as eggs enriched in n3 fatty acids, eggs from Green-legged partridge and eggs from partridge quail. It gives the overview of the vast majority of types of eggs available for the regular consumers in the European Union. Moreover, for each type of eggs, eight different brands was investigated in three replicates, as explained in ‘material and methods’ section which gives 5 x 8 x 3 + 16 = 136 samples investigated. We strongly disagree with the statement that our study material is not enough for a comprehensive presentation of the overall population.
R: In the materials and methods: What do you mean with “The content of saturated fatty acids (SFA) ranged from 32.8% (N3) to 36.2% (Q) of total fatty acids (TFA).”?
A: We would like to explain, that all abbreviations were explained when used for the first time (as in this sentence saturate fatty acids (SFA), total fatty acids (TFA)) and all codes for groups of study material were also explained in ‘material and methods’ section (lines 65-73). We kindly ask Reviewer 3 for more detailed explanation of what is not clear for them in this statement.
R: Why the authors just focused on the mentioned two elements?
A: The focus of the article is on the lipid components. However, we decided to include information on iron and zinc because, from a nutritional point of view, eggs in the diet can provide a large part of the requirement for these nutrients. We felt that information on these nutrients would enrich our work and be of interest to our readers. In the available scientific literature, these nutrients are not reported for all egg groups commonly available in the EU.
R: What will be the message the authors want to deliver from the overall discrimination? Several concerns are raised against integrating the different systems applicability like in case of your mentioned fish sources in your discussion.
A: We are aware of some concerns regarding obtaining of fish oil. Some ecologists suggest algae oil as a substitute of fish oil, but the fatty acids profile of fish and algae oil is different regarding EPA and DHA content. Moreover, as we investigated the market products, we did not have any impact on the source of oils used in hens diet to obtain n3 enriched eggs. We cannot be burdened with guilt for the source of fish for fish oil production.
R: I think the overall analysis are so simple and needs more efforts to have the conclusions that your Bird eggs samples are almost perfect sources of all the nutrients necessary for development. Even, development of what? Also, have the authors made a positive and negative controls or just chemical discrimination by chromatographic analysis? Where the physiological measurements that should clarify the authors’ points. The presentation of the data are so simple and no integrations among the presented data.
A: We would like to explained that methods which we used in our study are well known and widely used for similar analysis. Moreover, our study did not focused on the application of the most difficult or complicated analytical methods which are available. Simplicity and availability of analytical methods make our results easy to repeat and compare with the results of other authors. In our opinion, this is the advantage of this study. As emphasized above, no physiological studies were meant to be performed and described in this manuscript. We would like to ask for the explanation what Reviewer 3 claims as negative controls in this precise experiment.
Round 2
Reviewer 3 Report
Comments and Suggestions for Authors
Dear author, first of all, I would like to mention that I am not against the publication of your article. I am helping you to present your article in the best way that allows it to be published in Foods. I invite the authors to pay more attention to their study.
Comments for the authors: The authors still did not respond to my comments properly. The authors argued the reviewer's concern regarding the physiological impact. In the abstract and other sections, they mentioned the nutritional value that should be related to any other nutritional or physiological parameters. I suggest that the authors should work more to present that study in the best way to be considered in the Foods journal.
Author response: “We strongly disagree with the statement as experiment integration was not well explained and was rather convoluted. We would like to ask for withdraw of this statement or we would like to ask Reviewer for the explanation of their opinion based on the manuscript text. The focus of the article is on the lipid components. However, we decided to include information on iron and zinc because, from a nutritional point of view, eggs in the diet can provide a large part of the requirement for these nutrients. We felt that information on these nutrients would enrich our work and be of interest to our readers. In the available scientific literature, these nutrients are not reported for all egg groups commonly available in the EU.”
Reviewer response: Could the authors mention how they came to the integration between Iron and Fatty acids as an example? What is the relationship between these two experiments to let the authors just focus on their conclusions? The authors just focused on a few in vitro chemical measurements that related to the eggs from the different franchises. without any significant contribution to the physiological impact related to the human body or even the animal in vivo studies.
A: We are aware of some concerns regarding obtaining of fish oil. Some ecologists suggest algae oil as a substitute of fish oil, but the fatty acids profile of fish and algae oil is different regarding EPA and DHA content. Moreover, as we investigated the market products, we did not have any impact on the source of oils used in hens diet to obtain n3 enriched eggs. We cannot be burdened with guilt for the source of fish for fish oil production.
Comment: Still did not get the relationship to the study objectives.
A: We would like to explained that methods which we used in our study are well known and widely used for similar analysis. Moreover, our study did not focused on the application of the most difficult or complicated analytical methods which are available. Simplicity and availability of analytical methods make our results easy to repeat and compare with the results of other authors. In our opinion, this is the advantage of this study. As emphasized above, no physiological studies were meant to be performed and described in this manuscript. We would like to ask for the explanation what Reviewer 3 claims as negative controls in this precise experiment.
Comment: What the reviewer means with simplicity is that it could not come to a comprehensive novel conclusion.
Comments on the Quality of English LanguageModerate editing of English language required.
Author Response
Dear Reviewer
Thank you for your time in reviewing our article.
In our paper, we presented a clear objective for the study. The aim of our work was to compare the content of lipid components as well as iron and zinc levels in chicken and quail eggs commonly available on the market (in response to Your comment we have decided to remove the phrase "and their nutritional value" from the manuscript).
No article of which we are aware presents a compilation of results from all types of eggs. We want to inform consumers about the products that are on the market shelves so that they can make conscious choices.
Our article was properly planned and executed. The results were presented in a way that was unquestionable to the other reviewers. Conclusions were drawn solely based on the research results.
We have set out below detailed responses to each of the Reviewer's allegations.
R: The authors still did not respond to my comments properly. The authors argued the reviewer's concern regarding the physiological impact. In the abstract and other sections, they mentioned the nutritional value that should be related to any other nutritional or physiological parameters. I suggest that the authors should work more to present that study in the best way to be considered in the Foods journal.
A: In our work, we have investigated the levels of some essential nutrients, which toughener with other components create the nutritive value of investigated products. Nutrients requirements differ for different populations, regarding for example gender, age, physiological or pathological conditions.
Of course, the nutrients in eggs will have an effect on the body. However, we did not study this. We only calculated the lipid quality indices. These indices are commonly used to evaluate lipids in different food products. The indices have been developed based on several studies. They indicate the potential effects that fat can have on the body, but we have never claimed to have done anything more than calculate, present, and compare the quality indices of egg lipids with other products. These indices are not physiological parameters.
R: Could the authors mention how they came to the integration between Iron and Fatty acids as an example? What is the relationship between these two experiments to let the authors just focus on their conclusions?
A: We deny that we are suggesting a relationship between the fatty acids and minerals studied. These are two separate parameters that determine the nutritional value of the products tested. We felt that adding information on iron and zinc would enrich our work and be of value to readers.
Of course, we could limit the article to the fatty components. However, we believe that the additional information on iron and zinc enriches our work. Nowhere have we found a compilation of lipids components, zinc and iron content of all the most common egg types in the EU.
R: The authors just focused on a few in vitro chemical measurements that related to the eggs from the different franchises. without any significant contribution to the physiological impact related to the human body or even the animal in vivo studies.
A: Yes. In our study, we focus on a few chemical measurements related to the eggs from the different franchises. We evaluate market products on selected parameters. We do not understand the Reviewer's objections to the premise of our study. Not every study has to show the effect of food/nutrients on a living organism. Moreover, the effect in humans may be different taking into account for example polymorphism of genes (a good example is the metabolism of folates) and other factors. The premise of our study was simple, but that does not mean it was wrong. The results of our study provide a lot of valuable information.
R; Comment: Still did not get the relationship to the study objectives (In the context of enriching eggs with acids of the N3 PUFA).
A: As we have explained previously, we have researched products available on the market for regular consumers. The eggs that we categorised as N3 had a clear statement on the pack that the n3 family polyunsaturated fatty acid content had been increased compared to regular products. We do not know how the producers have tried to increase the n3 PUFA content. Such information is not provided to the consumers.
The aim of our study was to compare the levels of lipid components, iron and zinc in commercially available chicken and quail eggs.
N3 eggs are widely available on the UE market.
We did not find any information in the literature on the n-3 PUFA content of eggs on the market that have a declaration on the pack claiming a higher content of this ingredient. Instead, there are several published studies on the possibility of increasing the amount of these fatty acids in eggs by modifying the diet of laying hens. However, these studies have been carried out under controlled conditions. We wanted to evaluate a marketable product.
R: What the reviewer means with simplicity is that it could not come to a comprehensive novel conclusion.
A: The conclusions of our study are very clearly and concretely presented in the last chapter.
We could not find a study that compiled our chosen parameters for the nutritional value of eggs available on the European Union market. Our study provides a lot of valuable information for consumers who will be able to make conscious decisions about such products.
We hope that the above explanations are sufficient for the Reviewer. We are aware that the design of our study was simple. However, this does not mean that the results are not interesting. We believe that the information provided in the article will be valuable to readers.